# Secondary Reporting of G6PD Deficiency on Newborn Screening

**DOI:** 10.3390/ijns9020018

**Published:** 2023-03-27

**Authors:** Stephanie C. Hoang, Pamela Blumenschein, Margaret Lilley, Larissa Olshaski, Aisha Bruce, Nicola A. M. Wright, Ross Ridsdale, Susan Christian

**Affiliations:** 1Genetics & Genomics, Alberta Precision Laboratories, Edmonton, AB T6G 2H7, Canada; 2Department of Medical Genetics, University of Alberta, Edmonton, AB T6G 2H7, Canada; 3Pediatric Hematology, Stollery Children’s Hospital, Edmonton, AB T6G 2B7, Canada; 4Department of Pediatrics, Cumming School of Medicine, University of Calgary, Calgary, AB T3B 6A8, Canada

**Keywords:** newborn screening, galactosemia, galactose-6-phosphate dehydrogenase deficiency, G6PD deficiency

## Abstract

In April 2019, the Alberta Newborn Screening Program expanded to include screening for classic galactosemia using a two-tier screening approach. This approach secondarily identifies infants with glucose-6-phosphate dehydrogenase (G6PD) deficiency. The goals of this study were (i) to evaluate the performance of a two-tier galactosemia screening protocol, (ii) to explore the impact on and acceptability to families of reporting G6PD deficiency as a secondary finding, and (iii) assess the communication and follow-up process for positive G6PD deficiency screening results. The two-tiered galactosemia approach increased the positive predictive value (PPV) for galactosemia from 8% to 79%. An additional 119 positive newborn screen results were reported for G6PD deficiency with a PPV of 92%. The results show that there may be utility in reporting G6PD deficiency results. Most parents who participated in the study reported having some residual worry around the unexpected diagnosis; however, all thought it was helpful to know of their child’s diagnosis of G6PD deficiency. Finally, the communication process for reporting G6PD deficiency newborn screen results was determined to result in appropriate follow up of infants.

## 1. Introduction

On 1 April 2019, the Alberta Newborn Screening Program expanded to include screening for classic galactosemia using a two-tiered screening approach. Although classic galactosemia is the primary target condition, other variants of galactosemia may be identified based on enzyme activity. Diagnostic erythrocyte galactose-1-phosphate uridylyltransferase (GALT) activity is expected to be absent or barely detectable in classic galactosemia, between 1 and 15% in clinical variant galactosemia, and between 15 and 25% in Duarte variant galactosemia [1,2]. Infants with classic galactosemia benefit from early detection and immediate treatment to reduce the risk of serious complications including seizures and sepsis [1,3]. This requires urgent contact with the family by the clinical metabolic team to initiate confirmatory testing and follow-up management including assessing whether the infant should be immediately placed on a lactose-restricted diet. Similarly, a lactose-restricted diet for infants with clinical variant galactosemia prevents acute complications and appears to limit the risk for long-term complications [1]. In contrast, Carlock et al. found that infants with Duarte variant galactosemia remained asymptomatic and did not benefit from dietary intervention [4].

Alberta uses an opt-out approach for newborn screening (NBS). This approach is common among NBS programs due to the significant implications of an abnormal result. Over 99% of infants in Alberta undergo NBS [5]. Classic galactosemia screening in Alberta was modeled after the approach used in the Netherlands [6]. First-tier testing was performed using the Neonatal GALT kit from PerkinElmer Inc. [7] and measures the total enzyme activity in the following series of chemical reactions via formation of NADPH+ which is quantitated. First, galactose-1-phosphate is converted to glucose-1-phosphate by the GALT enzyme. Then, glucose-1-phosphate is dehydrogenated by phosphoglucomutase to produce glucose-6-phosphate. Glucose-6-phosphate dehydrogenase (G6PD) then converts glucose-6-phosphate to 6-phosphogluconate which is lastly converted to ribulose-5-phosphate by 6-phosphogluconate dehydrogenase. Deficiencies of any of the first three enzymes result in a reduced production of NADPH+ and an abnormal first-tier result. Infants with enzyme activity below the reference interval (normal >3.8 U/dL) have to total galactose (Tgal) quantitation. Infants with high Tgal levels (≥610 µmol/L if on a lactose diet and ≥64 µmol/L if on a lactose-free diet) screen positive for galactosemia, while infants with normal or decreased Tgal levels screen positive for G6PD deficiency. Tgal quantitation therefore decreases the number of false positive newborn screens for galactosemia and allows the secondary identification of G6PD deficiency. After consultations with stakeholders in metabolics and hematology, the lab decided to report the secondary finding of G6PD deficiency because of the potential benefits of early diagnosis and treatment.

The Netherlands approach [6] was modified by factoring in the infant’s diet as indicated on the requisition. Diet-based cut-offs were derived from the current understanding of expected Tgal values in the affected and normal populations [8,9]. Infants with galactosemia on a lactose diet have well-characterized elevations of Tgal, whereas affected patients on a lactose-free diet have reduced levels of Tgal. Therefore, it is expected, but has not been reported, that infants who are incidentally on a lactose-free diet at the time of NBS collection would have lower Tgal levels. The lab chose to utilize dietary-specific cutoffs to reduce the risk of false negative results.

Since the primary target of screening is classic galactosemia, this assay is not optimized to detect all cases of G6PD deficiency. Some infants with G6PD deficiency are expected to have enzyme activity above the screening cut-off and therefore have a normal NBS result. In addition, the first-tier assay lacks sensitivity to distinguish classic galactosemia, clinical variant galactosemia, and Duarte galactosemia. Therefore, patients with clinical variant or Duarte galactosemias may have a positive classic galactosemia NBS result.

Individuals with G6PD deficiency have an increased risk of developing neonatal hyperbilirubinemia which can lead to complications such as hemolytic anemia [10]. Hemolytic anemia results from oxidative stress when an individual is exposed to environmental triggers such as infection, moth balls, fava beans, and some commonly used medications. G6PD deficiency is an X-linked recessive condition with an incidence of 1/19 worldwide [11]. It is often seen in people of African, Asian, Mediterranean, and Middle Eastern descent. The World Health Organization recommends population screening in areas where the incidence in males is greater than 3–5% [12]. Therefore, some countries such as Singapore and the Philippines screen for G6PD deficiency as a primary target in NBS [13,14].

For positive classic galactosemia NBS results, families are contacted immediately by the clinical metabolic team to assess the infant and arrange diagnostic testing and treatment. In contrast, positive NBS results for G6PD deficiency are handled less urgently. The providers are notified of the results by the NBS lab within three business days. A G6PD deficiency information sheet is sent to the ordering provider which includes instructions regarding disclosing the NBS result, assessing the infant, counselling the family about avoiding oxidative stress triggers, ordering confirmatory, and other follow-up testing (i.e., bilirubin assessment), as well as sending a routine referral to a pediatric hematology specialist.

The goals of this study were (i) to evaluate the performance of a two-tiered galactosemia screening protocol, (ii) to explore the impact on and acceptability to families of reporting G6PD deficiency as a secondary finding, and (iii) to assess the communication and follow-up process for positive G6PD deficiency NBS results.

## 2. Methods

### 2.1. Chart Review

A retrospective chart review was designed to evaluate the performance of this two-tiered approach for galactosemia NBS. Outcome data on infants who screened positive for galactosemia or G6PD deficiency born between 1 April 2019 and 31 March 2022 were recorded. The follow-up testing for each infant was reviewed, and the positive predictive value (PPV) for galactosemia and G6PD deficiency were calculated. Galactosemia PPV was calculated with and without Duarte galactosemia and clinical variant galactosemia cases based on erythrocyte GALT activity (U/g) and molecular genetic testing results. The G6PD deficiency PPV was calculated based on G6PD activity (U/g Hb). In the G6PD deficiency group, bilirubin values were recorded to determine adherence to the recommended follow-up and the possible impact of reporting G6PD deficiency via NBS.

Data analysis was descriptive in nature. Continuous variables are presented as the mean and range. Categorical variables are presented as counts and percentages. This chart review received approval from the University of Alberta Health Research Ethics Board (Pro00120081). Individual consent was not required as the data were compiled and anonymous.

### 2.2. Family Survey

Parents of infants with a positive G6PD deficiency NBS result who were referred and seen by pediatric hematology specialists between 1 February 2021 and 31 July 2022 were invited to participate in the study. Parents consented to participate following a discussion about the diagnosis of G6PD deficiency with a hematology specialist and were asked to complete two surveys following their appointment. The first survey was emailed shortly after the appointment and evaluated their prior awareness of G6PD deficiency and their satisfaction with the notification process for G6PD deficiency NBS results. This survey was adapted from a survey in the study by Rueegg et al. assessing parents’ experience with a positive cystic fibrosis newborn screen result [15]. The second survey was sent to parents two weeks after their appointment with pediatric hematology and assessed their adaptation to the diagnosis of G6PD deficiency. This survey was adapted from the Psychological Adaptation to Genetic Information Scale (PAGIS) survey [16]. Parents of infants with false positive NBS results were excluded from this study. As an incentive, a USD 30 gift card was emailed to participants upon completion of the second survey. All survey responses were entered directly into REDCap (Research Electronic Data Capture) [17].

Due to the small sample size, data analysis of the survey results was descriptive in nature. To evaluate the participants’ adaptation to the diagnosis of G6PD deficiency for their infant, PAGIS scoring was calculated according to instructions provided by the survey creator (personal communication). In total, there were 26 items divided among five subscales. The subscales were non-intrusiveness, support, self-worth, certainty, and self-efficacy, which describe the components of psychological adaptation to genetic information. Each item was scored based on the participant’s response to the statement ranging from strongly agree to strongly disagree. The total PAGIS unweighted score could range from 26 to 156, with higher scores suggestive of better psychological adaptation. Participants scoring greater than 96 in this scale are considered well-adapted [18]. Both weighted and unweighted total scores were calculated to allow for comparison with other published studies. Continuous variables are presented as the mean and range. Categorical variables are presented as counts and percentages. Ethics approval was obtained from the University of Alberta Health Research Ethics Board (Pro00102518) and the University of Calgary Conjoint Health Research Ethics Board (REB20-2154).

## 3. Results

### 3.1. Chart Review

Between 1 April 2019 and 31 March 2022, a total of 148,425 infants had an NBS result reported in Alberta (99.37% of all registered infants). Of these, 133 infants had first-tier enzyme activity below the screening cut-off and had second-tier Tgal quantitation. Fourteen cases had a Tgal value greater than the appropriate diet-specific cut-off and were reported as positive galactosemia NBS cases. The remaining 119 infants had Tgal values below the appropriate cut-off and were reported as positive G6PD deficiency NBS cases.

The positive galactosemia results were reported when infants were on average 9 days of age (range 4–25 days) and are further described in Table 1. Biochemical and molecular diagnostic testing revealed that one infant had classic galactosemia, one had clinical variant galactosemia, and nine had Duarte variant galactosemia. The remaining three results were determined to be false positive screens based on diagnostic testing. The PPV with variant galactosemia (clinical and Duarte) included was 79%. The PPV for classic galactosemia, excluding variant galactosemia, was 7%.

A description of the screened positive G6PD deficiency cases is provided in Table 2. G6PD deficiency NBS results were reported when the infants were an average of 6 days old (range 2–18 days) and 78% (*n* = 93) attended a follow-up appointment with a pediatric hematologist by the completion of the study. Appointments with pediatric hematology occurred between 19 and 520 days of age with an average age of 82 days. Confirmatory quantitative G6PD testing was completed for 113 infants; the remaining 6 infants were lost to follow-up. This testing supported a diagnosis of G6PD deficiency for 104 infants, resulting in a PPV of 92% (*n* = 104/113). Enzyme activity in these patients was equal to or below the limit of detection of 0.7 U/g Hb, except for one case with an activity level of 1.0 U/g Hb. The normal reference range is 8.0–13.4 U/g Hb. There were nine false positive results.

Of all babies with a positive NBS result for G6PD deficiency, 91% (*n* = 108/119) had a bilirubin assessment performed. The bilirubin assessment was performed before the NBS was reported for 61% (*n* = 66/108) of infants and, after the NBS was reported, for 39% (*n* = 42/108) of infants. A reference range was provided for 64 initial bilirubin assessments (before NBS = 30; after NBS = 34). Of those performed before the NBS was reported, 53% (*n* = 16/30) were abnormal compared to 35% (*n* = 12/34) when the assessment was initiated after the NBS was reported.

### 3.2. Family Survey

During the recruitment period, 33 infants with a confirmed diagnosis of G6PD deficiency following a positive NBS were seen through one of the two pediatric hematology clinics in the province. Of these, 23 families were invited to complete the family surveys. Four families declined and three expressed interest but did not complete the survey. The participation rate for those invited was 70% (*n* = 16/23). The demographic data for the 16 respondents is shown in Table 3. Most study participants were the mother of the affected child (94%; *n* = 15) and of Filipino descent (94%; *n* = 15). Approximately half of the respondents had heard of G6PD deficiency prior to NBS (44%; *n* = 7) and/or knew someone who had G6PD deficiency (56%, *n* = 9).

Most of the respondents were first informed of their child’s positive NBS for G6PD deficiency by their child’s pediatrician (Table 4). Approximately half received the information during an appointment (43%; *n* = 7). The remainder were told over the phone (31%; *n* = 5) or at the hospital (25%; *n* = 4). Of those told in hospital, one study participant was informed before discharge following delivery, two were informed when the newborn was admitted for jaundice, and one did not specify. Most respondents felt that sufficient information about G6PD deficiency was provided by their child’s healthcare provider before meeting with the specialist (75%; *n* = 12). All respondents felt that enough information was provided to them about G6PD deficiency by the specialist. Four respondents (25%) indicated that the wait between the diagnosis and the appointment with the specialist was too long. However, the wait times among these individuals differed dramatically (<7 days to >6 months).

After receiving their child’s diagnosis, but before meeting with the specialist, 12 respondents (75%) reported being somewhat worried and 4 (25%) reported being very worried (Table 5). After meeting with the specialist, 4 (25%) reported being not at all worried, and 12 (75%) reported being somewhat worried. When asked if they thought it was helpful to know of their child’s diagnosis of G6PD deficiency, all respondents answered yes.

The average PAGIS total and subscale scores are reported in Table 5. On a scale of 1 to 6, with higher scores indicating better adaptation, mean scores ranged from 3.3 for the non-intrusiveness subscale to 4.9 for the support subscale. The mean total unweighted PAGIS score was 115.0 and the weighted PAGIS score was 22.0. Three quarters of respondents (*n* = 12/16) had an unweighted total PAGIS score above 96, which is considered well adapted [18]. Of the 4 respondents with scores <96 (poorly adapted), none knew someone with G6PD deficiency or had heard of the condition. Half (*n* = 2/4) of these respondents were very worried and half (*n* = 2/4) were somewhat worried before meeting with the specialist and all (*n* = 4/4) remained somewhat worried after meeting with the specialist. In comparison, of the 8 respondents with a total unweighted PAGIS score >96 (well adapted), 67% (*n* = 8/12) had heard of G6PD deficiency and 75% (*n* = 9/12) knew someone with it. Prior to meeting with a specialist, 17% (*n* = 2/12) were very worried and 83% (*n* = 10/12) were somewhat worried, whereas after meeting with the specialist, 67% (*n* = 8/12) were somewhat worried and 33% (*n* = 4/12) were not at all worried.

## 4. Discussion

The first goal of this study was to evaluate the performance of a two-tiered approach to NBS for classic galactosemia. Seven percent (*n* = 1/14) of infants with a positive NBS result had a confirmed diagnosis of classic galactosemia and 79% (*n* = 11/14) had a diagnosis of either classic or a variant form of galactosemia. Therefore, the incidence of classic galactosemia over our study period (April 2019 to March 2022) in Alberta is 1/148,425 (0.7/100,000), and the incidence of galactosemia (classic and clinical variant) benefiting from early detection and immediate treatment in the same period is 2/148,425 (or 1.34/100,000). If the second-tier test was not performed, the PPV would have been 0.8% (1/133) excluding variant galactosemias and 8% (11/133) including variant galactosemias. The PPV for classic galactosemia NBS was significantly improved by the addition of the second tier and 119 babies did not require clinical assessment for classic galactosemia which minimized stress on the families, avoided burdening metabolic specialist clinics unnecessarily, and saved costs associated with referral testing.

It is also important to appreciate the impact of reporting G6PD deficiency as a secondary finding. Bilirubin assessment data suggests that there may have been increased screening for hyperbilirubinemia after a positive G6PD deficiency NBS result. In total, 42 bilirubin assessments (39%) were performed after the NBS was reported and of the 34 results with a reference range provided, 12 (35%) were abnormal. We therefore speculate that reporting G6PD deficiency during NBS may have resulted in improved detection of neonatal hyperbilirubinemia and fewer complications of untreated hyperbilirubinemia. It is challenging to assess the medical benefits related to identifying hyperbilirubinemia due to the rarity of severe complications for individuals with G6PD deficiency. To assess whether hyperbilirubinemia is frequent and serious enough to justify NBS, its incidence in patients with G6PD deficiency in Canada needs to be determined as reviewed by Leong [19]. There is likely an increasing prevalence of this disorder in Canada due to immigration from populations known to be at-risk for G6PD deficiency such as those in the Philippines. In Alberta, Canada, there was a 35% increase in people who stated their place of birth was the Philippines from 2016 to 2021 census data [20,21]. Additional data regarding the incidence of G6PD deficiency, as well as large long-term studies looking at the frequency of severe outcomes, would be required for purposeful consideration of G6PD deficiency as a primary NBS target. Further, there are limitations to our screening approach since it was designed for classic galactosemia screening and does not have a 100% sensitivity for G6PD deficiency. Although we are unable to calculate the false negative rate of the assay, the results suggest that the most severe cases of G6PD deficiency were detected as indicated by the results of diagnostic G6PD testing.

From the family perspective, parental anxiety remained for the majority (75%) of parents following the disclosure of the G6PD deficiency NBS result. However, most parents adapted well to the confirmed diagnosis within a couple weeks of meeting with a specialist. Adaptation may be influenced by awareness of the condition, as none of the parents with unweighted total PAGIS scores consistent with poor adaptation had previously heard of G6PD deficiency or knew someone with it. A comparison with adaptation to other genetic diagnoses that have been evaluated using the PAGIS measure was variable. The average unweighted total PAGIS score in this study (115.0) was similar to those of families given a diagnosis of hypertrophic cardiomyopathy (114) [22]. Compared with patients with genetically confirmed catecholaminergic polymorphic ventricular tachycardia (CPVT), parents in our study had lower scores on the non-intrusiveness subscale (3.3 vs. 3.9) but higher scores for each of the other scales (support 4.9 vs. 4.0; self-worth 4.5 vs. 4.2; certainty 4.8 vs. 4.4; self-efficacy 4.6 vs. 4.0) [23]. G6PD deficiency has a much higher prevalence than CPVT and many of our study participants were previously aware of G6PD deficiency, perhaps partly explaining the higher scores on these subscales. In contrast, the average weighted total PAGIS score of 22.0 in our study was slightly lower than scores in a study by Read et al. of families with a variety of genetic diagnoses (24.3) [16]. Limited data were provided on the diagnoses included in the study by Read et al., making speculation about the difference challenging. Based on the findings in our study, providers may consider providing patient resources to families earlier in the follow-up process and providing additional reassurance regarding the expected long-term outcome for infants with this diagnosis. This approach is supported by the results of a recent systematic review of psychosocial issues related to NBS [24]. Overall, respondents in our study felt that they had sufficient information after meeting with the specialist (*n* = 16/16), showing that parents’ needs were met to adapt to this unexpected health information about their infant.

While the current study did not explore healthcare providers’ views of reporting G6PD deficiency as a secondary finding on NBS, a US study found that pediatric providers in Ohio supported G6PD screening. They reported that a neonatal diagnosis of G6PD deficiency was helpful in management and the diagnosis changed their practice including increased counselling about jaundice, avoidance of hemolytic triggers, and increased bilirubin testing [25]. These providers felt that the diagnosis was generally well accepted and understood by most parents which is consistent with the results from our study.

The final goal of our study was to assess the communication and follow-up process for positive G6PD deficiency NBS results. When notified of a positive NBS for G6PD deficiency, ordering providers are directed to order several diagnostic tests including G6PD quantitative testing and a bilirubin assessment, as well as to make a referral to a pediatric hematologist. Diagnostic G6PD deficiency testing occurred for 95% of infants with a positive G6PD deficiency NBS result (*n* = 113/119) and a bilirubin assessment was ordered for most infants (91%; *n* = 108). More than three quarters (78%; *n* = 93/119) of the families were referred to pediatric hematology specialists for further counselling and/or follow-up testing. In total, six (5%) infants were lost to follow-up meaning a diagnostic outcome could not be recorded. This is comparable to the number of infants lost to follow up for other screened disorders in Alberta [5]. These findings suggest that the reporting process used for G6PD deficiency NBS results is acceptable and appropriate follow-up of these infants occurred following notification of the positive NBS result.

The study limitations identified are as follows. The prevalence of bilirubin assessments was determined from a review of lab data for infants with a positive G6PD deficiency NBS result and so we were unable to determine the indication for each assessment. Our data also did not provide information on the prevalence of bilirubin assessments in our study group compared with the general neonatal population. The bulk of the recruitment period for the family survey branch of this study occurred during the height of the global COVID-19 pandemic. In addition, our study relied on pediatric hematology clinics with limited staff to identified potential study participants. The study team was not able to directly invite families to complete the family survey. Patient management was directly affected due to pandemic mitigation protocols and thus, study participation fell significantly short of our expectations. Although the number of false positive screens for G6PD was small (9/113 or 8%), excluding this sub-population could have biased our results, potentially overestimating the benefits of reporting G6PD deficiency NBS results. We did not expect that infants with a false positive G6PD deficiency NBS result would be referred to pediatric hematology specialists, so we excluded them from family survey participation. Ultimately, three out of the nine cases were referred to a hematology clinic. It is also important to note that the data analyzed are specific to the Alberta population, possibly limiting the generalizability of our findings.

## 5. Conclusions

In summary, our study was able to evaluate a two-tiered approach to classic galactosemia NBS which allowed for the reporting of G6PD deficiency results. The data suggest that there may be utility in reporting such results. Within 3 years, the use of Tgal saved 119 infants from a false positive investigation for galactosemia. Currently, G6PD deficiency screening as a primary target is performed in a small number of programs worldwide. Here, some cases of G6PD deficiency were detected with a high specificity as part of the two-tiered classic galactosemia screening method without additional lab costs. When we evaluated the impact such results have on families, we found that anxiety remained for most parents who participated in the study, although all thought it was helpful to know of their child’s diagnosis of G6PD deficiency. The secondary finding of G6PD deficiency during NBS may lead to increased identification of neonatal hyperbilirubinemia and may decrease the incidence of episodes of hemolysis in these patients, though further longitudinal study would be required. In addition, two-tiered classic galactosemia screening saved a significant number of families from the anxiety associated with the work-up required to rule out a potential galactosemia diagnosis.

## Figures and Tables

**Table 1 IJNS-09-00018-t001:** Positive galactosemia newborn screening and following diagnostic results in Alberta between April 2019 and March 2022.

Characteristic	*n* = 14	%
Sex		
Males	7	50
Females	7	50
Age at report of NBS results	9.2 days (range 4–25 days)	
Diagnosis		
Confirmed classic galactosemia	1	7
Confirmed clinical variant galactosemia	1	7
Confirmed Duarte variant galactosemia	9	64
False positive	3	21
Age at confirmation of diagnosis	18.5 days (range 5–63 days)	

**Table 2 IJNS-09-00018-t002:** Positive G6PD deficiency newborn screening and following diagnostic results in Alberta between April 2019 and March 2022.

Characteristic	*n* = 119	%
Sex		
Males	106	89
Females	13	11
Age at report of NBS results	6 days (range 2–18 days)	
Diagnosis		
Confirmed G6PD deficiency	104	87
False positive	9	8
Males	3	
Females	6	
Lost to follow up	6	5
Age at confirmation of diagnosis	58 days (range 1–371 days)	
Follow-up with Pediatric Hematology		
Edmonton clinic	65	55
Calgary clinic	28	24
Not seen in Pediatric Hematology	26	22
Age at consult with Pediatric Hematology	82 days (range 19–520 days)	
Bilirubin screening performed		
Before NBS reported	66	55
After NBS reported	42	35
Not performed	11	9

**Table 3 IJNS-09-00018-t003:** Family survey responses to demographic questions.

Question	*n* = 16	%
Respondents’ self-reported ethnicity		
Asian	1	6
Filipino	15	94
Is your child with recently confirmed G6PD deficiency your first-born child?		
Yes	6	37.5
No	10	62.5
Respondent’s self-reported relationship to child		
Mother	15	94
Father	1	6
How long have you been living in Canada?		
Since birth	2	13
Not since birth	14	88
Had you heard of G6PD deficiency before newborn screening?		
Yes	7	44
No	9	56
Do you know someone who has G6PD deficiency?		
Yes	9	56
Family member diagnosed with G6PD deficiency	7	
Friend diagnosed with G6PD deficiency	2	
No	7	44

**Table 4 IJNS-09-00018-t004:** Family survey responses to assess initial communication and specialist communication.

Question	*n* = 16	%
Who told you about the G6PD deficiency newborn screen result for your baby?		
My child’s pediatrician	11	69
My family doctor	2	13
My midwife	0	0
Other	3	19
How were you told about the G6PD deficiency newborn screen result for your baby?		
At an appointment with my baby’s healthcare provider	7	44
By telephone	5	31
At the hospital	4	25
Was enough information about G6PD deficiency provided by your baby’s healthcare provider before your appointment with the specialist?		
Yes	12	75
No	4	25
How long was it between when you were told about the G6PD deficiency newborn screen result for your baby and the appointment with the specialist?		
7 days	3	19
7 days–1 month	7	44
1–3 months	3	19
4–6 months	1	6
>6 months	2	13
Was the time between diagnosis and the appointment with the specialist reasonable?		
It was just right	11	69
It was too short	1	6
It was too long	4	25
Was enough information provided to you about G6PD deficiency by the specialist?		
Yes	16	100
Do you think it is helpful to know that your child has G6PD deficiency?		
Yes	16	100

**Table 5 IJNS-09-00018-t005:** Family survey responses to the adapted PAGIS survey.

Subscale	Scale Range	ME	Participant Range
Non-intrusiveness	1.0–6.0	3.3	1.2–5.0
Support	1.0–6.0	4.9	3.2–6.0
Self-worth	1.0–6.0	4.5	1.5–6.0
Certainty	1.0–6.0	4.8	3.8–5.8
Self-efficacy	1.0–6.0	4.6	3.0–6.0
Total unweighted PAGIS score	26.0–156.0	115.0	77.0–143.0
Total weighted PAGIS score	5.0–30.0	22.0	14.8–27.6

## Data Availability

The data presented in this study are available in Appendix A: Outcome from positive newborn screening for galactosemia and Appendix A: Outcome from positive newborn screening for G6PD deficiency.

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
