# Peer review of "Secondary Reporting of G6PD Deficiency on Newborn Screening"

_2409-515X, 2023, doi:10.3390/ijns9020018_

Round 1

Reviewer 1 Report

Dear authors,

You present an interesting study related to the value of a two tier approach for the screening for galactosemia(s) and in addition, the detection of another condition, G6PD deficiency, via the screening for galactosemia(s). The value of the two-tier approach for galactosemia screening is confirmed. The manuscript also describes both the screening aspects of and the impact on the families of children referred for G6PD deficiency as incidental finding of the newborn screening.

I would be grateful you could address the following comments that I have:

Line 29/30: what is the target condition for your programme? Classic galactosemia or Classic galactosemia and the other variants? Accurate assessment of the current screening programme can only be made when the target condition is described.

Line 43/44: could the authors elaborate on how the diet specific cut-off values are determined for their target condition(s)? The two-tier approach is based on the Dutch NBS algorithm, however the Dutch NBS programme does not apply diet specific cut-offs for TGAL; this should be specified.

Line 59/60: could the authors please describe how neonates were included in this study? Was informed consent obtained from the parents? If yes, how was that done and what was the rate of participation? If informed consent was not obtained prior to inclusion of a neonate in the study, could the authors please describe the ethical considerations/ethic committee evaluations that have been made to enable this study.

Line 100/101: could the authors please elaborate on why the parents of infants with a false positive result are excluded from the study? Could the authors explain why they regard the impact of a false positive referral for an 'incidental' as not informative/important for the evaluation.  Can you please comment on whether these exclusions could introduce a bias into the evaluation (parents of children with the condition are likely to be more positively impacted by this study than parents of children with a false postive screen and who have undergone unnecessary follow-up)?

Line 148/149: Of the group neonates that received a bilirubin assessment prior to the newborn screening: what was the reason for this and did all of these neonate have a positive G6PD result? 

Line 161: please correct the English. It should be '...were the mother of THE affected child...'

Line 181-186: Impact of this screening is described: 100% express worry before meeting with specialist; after meeting with specialist 75% remain worried to some extent. There is clearly an impact on these families following diagnosis. Please link these findings with the elaboration of the set-up of the study: were families recruited using informed consent? If so, what information did the parents receive about G6PD prior to participation in the study? Could these 'negative' impacts of the screening be mitigated by provision of more information prior to participation in the study?

Line 226: Can the authors please comment of the proportion of births by immigrant mothers in Canada? Could this in itself justify a primary G6PD screen?

Line 283/284: Please could the authors describe what is meant by 'arm's length approach to participation'? As written in the method it appears that this study has been conducted without review of an ethics committee and without informed consent of the parents. 

5. Conclusions: It is clear that the value/benefit of the two tier approach, established by the Dutch newborn screening programme, has been confirmed by this study. With respect to the impact of detection of G6PD as an incidental finding: alongside gratitude that their child has received a diagnosis, 75% parents express worry after consultation with a medical specialist. This impact on the families has not been addressed in this manuscript and as such the negative impact of reporting the secondary finding has been underestimated

Author Response

Thank you for your thoughtful comments. Please see the attachment.

Reviewer 2 Report

This is an interesting paper and combines descriptive results with the impact of a serendipitously ascertained secondary abnormality. I have a few, but I believe important comments.

1. a minor comment concerns a statement in the introduction. G6PD deficiency is described as an X-linked dominant condition. It is in fact, co-dominant except in the hemizygous circumstance. 

2. You must present the method to explain why G6PD deficiency is picked up incidentally. It is possible that a part of the screening community will know this, but papers are not written for a small cognoscenti, but rather should be understood by a larger audience. It is p[ossible to screen for galactose-1-phosphate uridyltransferase without picking up G6PD deficiency incidentally.

3. G6PD deficiency is important enough to be screen in high incident countries, but has never fallen into this category in North America or northern Europe. The justification for screening should be discussed. Is this just a fortunate outcome of the methods they use, or should it be done more widely? I come from California with a lot of people of SE Asian background. Should we be screening for it? Should people in the South be screening for it because of the large number of people of African background?

4. What is clinical variant galactosemia that has to be treated? This should be defined. I am a pretty experienced inborn error person and I am unsure what the term means.

Author Response

(The authors gave the same response as above.)

Round 2

Reviewer 1 Report

Here are my comments: Line 30: add the word 'condition' ('.....primary target condition.....') Line 43: Despite opt-out recruitment approach, I have concerns regarding the ethics of adding a condition to a screening panel (as routine or pilot), without informing the parents. Prior to screening is information made available to the parents via a website, pamphlet or other means so that they can make an informed decision? The authors have clarified that approval was made for this study by an ethics board (additions in lines 146-148), which is most important in relation to this manuscript (and therefore no further action is required). Further, my other comments and queries have been addressed by the authors by addition/amend of the manuscript that was originally submitted. With these changes incorporated I feel that this manuscript can now be accepted for publication.

Author Response

Thank you for your comments. We appreciate your review.